

# The effect of drought on dissolved organic carbon (DOC) release from peatland soil and vegetation sources

Jonathan P. Ritson[1], Richard E. Brazier[2], Nigel J.D. Graham[1], Chris Freeman[3], Michael R. Templeton[1] and Joanna M. Clark[4].

[1] Department of Civil and Environmental Engineering, Imperial College London, South Kensington, London, SW7 2AZ, UK

[2] Geography, College of Life and Environmental Sciences, University of Exeter, EX4 4RJ, UK

[3] Wolfson Carbon Capture Laboratory, School of Biological Sciences, Bangor University, Bangor, Gwynedd, LL57 2UW, UK

[4] Department of Geography and Environmental Science; School of Archaeology, Geography and Environmental Science; The University of Reading, Whiteknights campus, PO Box 227, Reading, RG6 6AB, UK.

*Correspondence to*: Jonathan Ritson (j.ritson12@imprerial.ac.uk)

**Abstract:** Drought conditions are expected to increase in frequency and severity as the climate changes, representing a threat to carbon sequestered in peat soils. Downstream water treatment works are also at risk of regulatory compliance failures and higher treatment costs due to the increase in riverine dissolved organic carbon (DOC) often observed after droughts. More frequent droughts may also shift dominant vegetation in peatlands from *Sphagnum* moss to more drought tolerant species. This paper examines the impact of drought on the production and treatability of DOC from four vegetation litters (*Calluna vulgaris*, *Juncus effusus*, *Molinia caerulea* and *Sphagnum spp.*) and a peat soil. We found that mild droughts caused a 39.6% increase in DOC production from peat and that this DOC was harder to remove by conventional water treatment processes (coagulation/flocculation). Drought had no effect on DOC production from vegetation litters, however large variation was observed between typical peatland species (*Sphagnum* and *Calluna*) and drought tolerant grassland species (*Juncus* and *Molinia*), with the latter producing more DOC per unit weight. This would therefore suggest the increase in riverine DOC often observed post-drought is due entirely to soil microbial processes and DOC solubility rather than litter-layer effects. Long term shifts in species diversity may, therefore, be the most important impact of drought on litter layer DOC flux, whereas more immediate effects are observed in peat soils. These results provide evidence in support of catchment management which increases the resilience of peat soils to drought, such as ditch-blocking to raise water-tables.

**Keywords:** Dissolved organic carbon, DOC, drought, peat, drinking water treatment

## 1.0 Introduction

Organic rich peat soils are a major global carbon sink (Limpens et al., 2008) which have formed due to the limited decay of recalcitrant plant litter found in peatland areas, coupled with anoxic conditions created by high





water-tables slowing decay (Billett et al., 2010; van Breemen, 1995). The locations in which these conditions
exist are threatened by climate change (Clark et al., 2010; Gallego-Sala and Prentice, 2012), and future climate
may also destabilise sequestered carbon (Evans and Warburton, 2010; Fenner and Freeman, 2011; Freeman et
al., 2001a).
Dissolved organic carbon (DOC) represents a significant flux of carbon from peatlands (Dinsmore et al., 2010)
and can also lead to difficulties for downstream drinking water treatment plants. DOC can cause colour, odour
and taste problems in drinking water and so must be removed as best as possible during treatment, commonly by
coagulation, flocculation and sedimentation/flotation. Any DOC which remains may act as a substrate for
microbial growth in the distribution system (Rodriguez and Sérodes, 2001) and can react during disinfection to
form disinfection by-products (DBPs) (Rook, 1974) which may have human health implications due to their
potential genotoxicity and carcinogenicity (Nieuwenhuijsen et al., 2009).
Droughts are projected to become more common under future climate conditions in the UK (Jenkins et al.,
2009). Droughts can have drastic consequences for peatland carbon storage and riverine DOC concentrations
due to the 'enzymatic latch' mechanism, whereby decomposition is supressed due to the inhibitory effect of
phenolic compounds. Under drought conditions, the water table is lowered, creating oxic conditions which
stimulates phenol oxidase enzymes, thereby reducing the concentration of phenolics and their inhibitory effect
on hydrolase enzymes (Fenner and Freeman, 2011; Freeman et al., 2001a). Altered redox conditions can also
change the controls on DOC solubility, meaning organic carbon is not solubilised during the drought but instead
flushed from the system once redox conditions return to normal (Clark et al., 2006, 2005; Clark et al., 2011).
These processes have led to numerous observations of increased riverine DOC after droughts which may remain
elevated for years after the event (Evans et al., 2005; Scott et al., 1998; Watts et al., 2001; Worrall and Burt,
2004). How drought effects the treatability of DOC is less well understood although some authors have noted an
increase in the hydrophilic component during droughts and more hydrophobic character post-drought (Clark et
al., 2011; Scott et al., 1998; Watts et al., 2001). Hydrophobic DOC is commonly regarded as being easier to
remove via coagulation than the hydrophilic fraction (Bond et al., 2011; Matilainen et al., 2010).
The impact of climate change on DOC production and drinking water treatment is complex and involves a
number of biogeochemical cycles (Ritson et al., 2014b). Vegetative change in peatlands has occurred in the
recent past (Chambers et al., 2007b) and is projected to continue with *Sphagnum* mosses, which are favoured for
peat formation, giving way to vascular plants (Fenner et al., 2007; Weltzin et al., 2003). Many grassland species
(*Juncus effusus*, *Molinia caerulea*) have encroached on peatland areas as a result of anthropogenic pressures
such as nutrient deposition and management practices (Berendse, 1994; Chambers et al., 2007a; McCorry and
Renou, 2003; Shaw et al., 1996). These species are adapted to higher nutrient availability (Aerts, 1999) and thus
can out-compete peatland species if nutrient levels are elevated through, for example, nitrogen deposition
(Berendse et al., 2001).
Vegetative change has implications for carbon storage in peatlands, as *Sphagnum* is responsible for a number of
mechanisms (e.g. the production of recalcitrant litter) which allow carbon to be stored over long time periods
(van Breemen, 1995). Conversely, many vascular plants can destabilise colonised peat, stimulating
decomposition by adding labile carbon at the surface and through their root systems (Fenner et al. 2007; Gogo et
al. 2010). As such, a number of programmes have aimed to promote *Sphagnum* dominance for carbon storage



and other ecosystem services (Grand-Clement et al., 2013). However, further evidence is needed on the water
quality outcomes of such interventions and the implications for water treatment.
Previous work has highlighted both the vegetative source and climate controls on production affecting the ease
of removal of DOC and the formation of DBPs (Gough et al., 2012; Reckhow et al., 2007; Ritson et al., 2014a;
Tang et al., 2013). The present research sought to quantify the effect of drought on peatland DOC flux and any
interaction with projected changes in litter input. To this end, climate simulations of varying drought severities
defined in terms of percentiles of mean monthly rainfall were performed on four typical peatland vegetation
types (*Calluna vulgaris*, *Juncus effusus*, *Molinia caerulea* and *Sphagnum spp.*) and a peat soil. After a six-week
drought simulation, the DOC released upon rewetting was analysed in terms of optical properties and
coagulation removal efficiency with ferric sulphate to determine: (a) whether drought conditions affect DOC
production from peatland litter and soil types and (b) whether peatland species and invasive, drought tolerant
vegetation produce different quantities and quality of DOC with respect to drinking water treatment.

**2.0 Methodology**
**2.1 Field site and sample collection**
Samples were collected from the Spooners site (51º 07'23.3'' N 3º 45'11.8'' W) in Exmoor National Park, UK at
approximately 400 m elevation. Further site details can be found in Ritson et al., (2014a). The site is part of the
MIRES project (Arnott, 2010) and was chosen as this area has been highlighted as a marginal peatland which
may be vulnerable to climate change (Clark et al., 2010).
Samples of vegetation and peat soil were collected in one day in May 2014 and were sealed in airtight bags in a
chilled container for transport from the field and stored in the dark at 4ºC before use. For vascular plants, litter
was collected as standing dead biomass. As the decomposition of *Sphagnum* is a continuum process, the section
2-4 cm below the capitulum was taken as equivalent to freshly senesced "litter", as in other studies (e.g.
Bragazza *et al*., 2007). Samples were sorted to remove any vegetation not belonging to the target species and
then cut to 2 cm length and homogenised. Peat samples were collected using a screw auger and peat from 10-30
cm depth was used in the experiments. Peat samples were sorted to remove as many roots as possible but in sites
where *Molinia* was present some fine roots remained.
The start times of the drought simulations for different DOC sources were staggered by up to two weeks to
allow prompt analysis of water extracts at the end of the experiments. Preliminary work suggested chilled
storage gave no significant difference in the amount of water extractable DOC or UV absorbance properties
after three weeks of storage in the dark at 4ºC.

**2.2 Experimental Design**
The vegetation and peat samples were homogenised by hand and randomly assigned a drought treatment in a
five (vegetation types) x four (drought treatments) design with five replicates per treatment, giving 100 samples
in total.
Data were obtained from regional historic climate records of the UK Meteorological Office for the south west of
England for the period 1910-2013 (UK Met Office 2014) and these values were used to define three severities of
drought and a control value. Data for the months of June, July and August (310 months in total) were used to



find the 50[th], 25[th], 10[th] and 5[th] percentile for total monthly rainfall and these values (Table 1) have been used to
set control, mild, moderate and severe droughts, respectively.

**Table 1: Monthly rainfall for control group and three severities of drought**

| Drought Treatment | Monthly rainfall total (mm) |
|---|---|
| Control (50[th] percentile) | 79.0 |
| Mild (25[th] percentile) | 51.5 |
| Moderate (10[th] percentile) | 34.7 |
| Severe (5[th] percentile) | 23.3 |


The number of days of rain per month was fixed at a baseline value of eleven (regional average for June, July
and August) and temperature ranged between the mean daily maximum of 18.9 for twelve hours and then and
the mean daily minimum of 10.7 °C for twelve hours, calculated using the same historical UK Meteorological
Office datasets for the south west of England.

**2.3 Experimental procedure and laboratory methods**
As in other decomposition studies, vegetation samples were air-dried to constant weight then mixed before
subsampling (e.g. Latter et al., 1998). Five subsamples of each vegetation type were then oven-dried at 70 °C
until constant weight, to determine the air-dry to oven-dry conversion factor. The peat samples were not air-
dried before use as this would have changed the redox conditions within the peat and created a hydrophobic
layer which can cause problems for re-wetting (Worrall et al., 2003). This will mean less accuracy in
determining the starting weight of the peat sample as some variation in water content may exist, however this
was minimised by effective homogenisation.
Buchner funnels fitted into amber-glass bottles were used to hold the sample and collect the simulated rainfall.
Approximately 2 g dry-weight of air-dried vegetation/peat was used, however a lower weight of sample was
used for *Sphagnum* (~0.65 g) and *Molinia* (~1.5 g) as this was enough to fill the Buchner funnel. The peat
samples were spread over the area of the funnel so that a seal was created and the simulated rainwater infiltrated
the peat rather than draining directly into the funnel.
The samples were then placed in an incubator for six weeks with simulated rainfall appliedeleven times per
month using high purity reverse osmosis (RO) treated water as per Table 1, following the methodology of
Ritson et al. (2016).
As the samples were collected from the field and had been in contact with litter and soil, no inoculation with
microorganisms was required as a suitable decomposer community was likely to be present (Van Meeteren et
al., 2007). In this experiment the action of invertebrates and other microfauna was excluded, however their role
in the decay of peatland litter is minimal (Dickinson and Maggs, 1974), although their role in DOC production
from peat soils may be more significant (Cole et al., 2002).
At the end of the six week simulation the samples were air-dried and weighed. Water extractable DOC from the
air dried sample was taken to simulate re-wetting following the end of the drought. DOC was extracted from soil
and vegetation samples using approximately 20:1 ratio of RO treated water to sample. Previous work has shown
that the amount of water used to extract DOC and whether one extraction is performed or sequential extractions





to simulate multiple rainfall events gives no significant variation in DOC quality (Don and Kalbtiz, 2005, Soong
et al., 2014), only changes in the total amount of carbon. DOC was measured as non-purgeable organic carbon
(NPOC) via a UV/persulphate oxidation method on a Shimadzu TOC-V instrument. The method detection limit
was determined by running five blank samples and using the value of three times the standard deviation. This
was found to be 0.05 mg $^{-1}$.
UV and fluorescence analysis was undertaken before coagulation/flocculation jar testing. UV absorbance was
measured on a Perkin Elma Lambda 3 using a 1-cm pathlength quartz cuvette and the specific absorbance,
SUVA, was calculated as the absorbance at 254 nm in units of m$^{-1}$ divided by the NPOC content (mgC l$^{-1}$).
Fluorescence analysis was completed using a Vary Eclipse fluorescence spectrophotometer where samples were
scanned at excitation wavelengths between 220 and 450 nm at 5 nm intervals and the resulting emission
recorded between 300 and 600 nm at 2 nm intervals. An R script was produced based on exiting scripts
(Lapworth and Kinniburgh, 2009) which performed a blank subtraction, masked out Rayleigh and Raman
scattering, visualised the data and calculated fluorescence indices. Data were normalised to the Raman
scattering peak of a RO water sample to allow comparison to other laboratories (Lawaetz and Stedmon, 2009).
The 'peak C' measure, related to humic-like character, and the tryptophan-like peak, 'peak T' were defined as in
Beggs et al., (2013).
Coagulation was performed on 350 ml of sample diluted to 3 mg l$^{-1}$ DOC using a Phipps and Bird PB-700
paddled jar-tester (Phipps and Bird Ltd., Virginia, USA). After settling, the sample was filtered by Whatman
qualitative grade 2 filters to remove flocs before NPOC analysis. Preliminary work indicated the following
conditions gave effective DOC removal of similar samples: pH 5.5, 30.0 mg l$^{-1}$ ferric sulphate dosed with 28.5
mg l$^{-1}$ calcium hydroxide for pH control during a flash mix of one minute at 175 rpm, followed by a slow mix of
30 minutes at 60 rpm and then one hour of settling. Assessment of DBP formation was attempted, however
analysis within the two week period specified in the method was not possible due to instrument failure so data
quality could not be assured.

**2.4 Data analysis and statistical methods**

Statistical analysis was performed in the open source programming language, R, and SPSS version 21 (IBM).
Due to problems with normality and heteroscedasticity a Box-Cox transform (Box and Cox, 1964) was applied
to the variables before testing with a factorial ANOVA. A Tukey HSD post-hoc procedure was used for
pairwise comparisons between the DOC sources and drought conditions. Estimates of effect sizes were made
using $\omega^2$ as this is suitable for small samples sizes (Keselman, 1975). Interactive effects from the omnibus
ANOVA were followed up using multiple one-way ANOVAs with a Holm-Šidák correction to control the
inflation of type one error (Holm, 1979; Šidák, 1967).

**2.5 Repetition of the control group**

To further investigate the effect of oxygenation of peat on DOC production and treatability, the control
condition of this experiment was repeated in August 2015 using peat samples collected from similar
ombrotrophic peatland sites in Dartmoor National Park (site details available in Ritson et al., 2016). Water
extractable DOC was taken from a subsample before the climate simulation began and analysed for fluorescence
and UV properties. Approximately 3.5 g dry weight of peat was then incubated using the same temperature and





rainfall as the control samples of the drought experiment with three replicates. After six weeks water extractable
DOC was again taken for fluorescence and UV analysis to assess any changes in DOC quality.

**3.0 Results**
**3.1 Omnibus ANOVA**
A factorial ANOVA was performed exploring the source, drought and interactive effects on DOC, SUVA, DOC
removal efficiency and the removal of SUVA (Table 2). Extractable DOC and SUVA had significant source,
drought and source*drought effects suggesting that there is variation in the sensitivity of the sources to drought.
No drought effects were observed for DOC removal or SUVA removal, although the source had strong effects
on these parameters. For all significant results the effect size for the source was much greater than that for the
drought treatment.

**Table 2: p-values from factorial ANOVA (significant values have been highlighted in bold and displayed**
**with $\omega^2$ estimate of effect size in brackets)**

| Variable<br><br>Factor | Water extractable<br>DOC | SUVA | DOC<br>removal | SUVA<br>removal |
|---|---|---|---|---|
| **DOC source** | **<0.001**<br>**(0.945)** | **<0.001**<br>**(0.422)** | **<0.001**<br>**(0.396)** | **<0.001**<br>**(0.331)** |
| **Drought** | **0.007**<br>**(0.004)** | **0.007**<br>**(0.034)** | 0.418 | 0.475 |
| **DOC source*Drought** | **0.050**<br>**(0.004)** | **0.005**<br>**(0.054)** | 0.234 | 0.951 |


**3.2 Water extractable DOC**
The omnibus ANOVA suggests both significant source and drought effects as well as an interaction, suggesting
the effect of drought varies between the sources. The mean DOC extracted for all samples from each source is
shown in Figure 1. The vegetation samples produced more DOC than the peat soil ($0.58 \pm 0.02$ mg g$^{-1}$) with the
peatland species, *Sphagnum* and *Calluna*, producing $3.47 \pm 0.30$ and $6.86 \pm 0.37$ mg g$^{-1}$, respectively whereas
the grassland species, *Juncus* and *Molinia*, produced much more at $9.21 \pm 0.62$ and $16.52 \pm 1.17$ mg g$^{-1}$,
respectively. A Tukey HSD test suggested that all DOC sources have significantly different means at the p<0.01
level except the *Calluna - Juncus* comparison which was significantly different at the p<0.05 level.




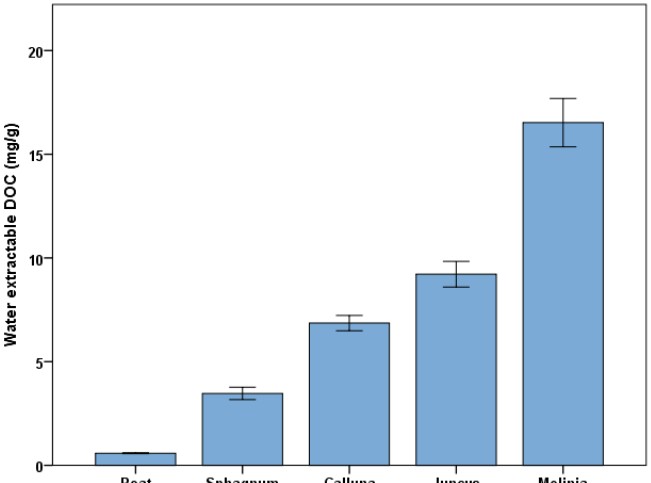


**Figure 1: Water extractable DOC of all samples across the different DOC sources (n=20 per source).**

**Error bars at one standard error.**


To investigate the source*drought interaction one-way ANOVAs were performed for drought effects on each of
the sources (Table 3) using a Holm-Šidák correction to control the inflation of type one error. This method
changes the value used for alpha, the significance level, based on how many comparisons have been performed
starting with the source with lowest p value and moving to the next lowest until an insignificant comparison is
found.

**Table 3: ANOVA results testing the effect of drought on water extractable DOC from different sources.**
**Significant effects (Holm-Šidák correction) are highlighted in bold with the $\omega^2$ estimate of effect size in**
**brackets.**

| DOC Source | p value (DOC extraction) | Alpha used for comparison |
|---|---|---|
| **Peat** | **0.010 (0.393)** | 0.010 |
| *Juncus* | 0.038 | 0.013 |
| *Sphagnum* | 0.097 | - |
| *Calluna* | 0.418 | - |
| *Molinia* | 0.550 | - |


Due to the decrease in the level of significance of the p value in the Holm-Šidák method only the peat source
was found to have a drought effect on water extractable DOC. The mean values were 0.48, 0.67, 0.61 and 0.58
mg g$^{-1}$ for the control, mild, moderate and severe treatments, respectively, and this is shown in Figure 2. The
mild drought treatment gave a significant increase in extractable DOC, indicated by a Tukey test for comparison
to the control group (p=0.007). This corresponded to a 39.6% increase in DOC production for the mild drought





treatment. Taken together, the main effects and interaction and $\omega^2$ values suggest that the source of DOC is the
most important factor on extractable DOC and that the effect of drought is significant only for the peat soil and
not for the vegetation.





**Figure 2: DOC extracted from peat on rewetting following different severities of drought (n=5 per**
**treatment). Letters indicate statistically similar groups from the Tukey test. Error bars at one standard**
**error.**

A larger standard error in the moderate and severe drought treatments meant that these were not significantly
different from the control (p=0.060 and p=0.204, respectively). Observations made throughout the experiment
suggested that in the severe treatment there was a large variation in the extent to which each replicate dried out.
Once peat becomes dry, a hydrophobic layer forms (Spaccini et al. 2002; Worrall et al. 2003), meaning that less
water will infiltrate the sample, therefore possibly increasing the severity of the drought beyond the
experimental design.
Variation in peat water content during the experiment was not recorded; however the water content of the peat
samples was measured at the end of the experiment. This averaged 16.11, 14.14, 15.11 and 5.95 g with standard
errors of 7.7, 3.0, 15.9 and 28.1% for the peat control, mild, medium and severe drought treatments respectively.
The much larger standard error in final water content agrees with observations during the experiment and could
perhaps explain some of the increased variation in extractable DOC for the severe drought treatment. This
hypothesis was tested by comparing the variation from group mean in final water content for each sample with
the variation from group mean in extractable DOC. These two measures of variance were found to correlate
(Spearman's ρ coefficient 0.484, p=0.031) suggesting some of the variation in DOC extracted may be explained
by different water contents between the samples in each treatment. This could have been caused by small
variations in the way rain was applied over the area of the sample or because shrinkage of the peat mass allowed





water to pass through the funnel rather than infiltrate the peat, again possibly increasing the severity of drought
beyond the experimental design.

**3.3 SUVA**
Mean values of SUVA in L mg$^{-1}$ m$^{-1}$ for the different sources were in the order *Molinia* (3.03 ± 0.38), peat (3.01
± 0.15), *Juncus* (2.04 ± 0.06), *Calluna* (1.66 ± 0.14) and then *Sphagnum* (1.34 ± 0.13). The Tukey HSD test
suggested that the mean values for SUVA formed three subsets with peat and *Molinia* > group two *Calluna* and
*Juncus* > *Calluna* and *Sphagnum*.

To investigate the source*drought interaction one-way ANOVAs were performed for drought effects on SUVA
from each of the sources (Table 4) using a Holm-Šidák correction.

**Table 4: ANOVA results testing the effect of drought on SUVA for different DOC sources. Significant**
**effects (Holm-Šidák correction) are highlighted in bold with the ω$^2$ estimate of effect size in brackets**

| DOC Source | p value (SUVA) | Alpha used for comparison |
|---|---|---|
| *Molinia* | **0.001 (0.546)** | 0.010 |
| *Sphagnum* | 0.278 | 0.013 |
| *Calluna* | 0.436 | - |
| Peat | 0.696 | - |
| *Juncus* | 0.741 | - |


Tukey's test suggested that both the moderate and severe drought treatments were significantly different than
the control (p=0.045 and 0.026, respectively) with means of 2.15, 4.09 and 4.27 L mg$^{-1}$ m$^{-1}$ for the control,
medium and severe treatment, respectively. Figure 3 shows a graph of SUVA for *Molinia* DOC from the
different treatment groups. The SUVA value approximately doubles between the control and the moderate and
severe droughts suggesting a large climatic control on the production of aromatic DOC from *Molinia* litter.
Taken together, the main effects and interaction and ω$^2$ values suggest that the source of DOC is the most
important factor on SUVA and that the effect of drought is significant only for *Molinia* litter and not for the
other vegetation types or the peat soil.




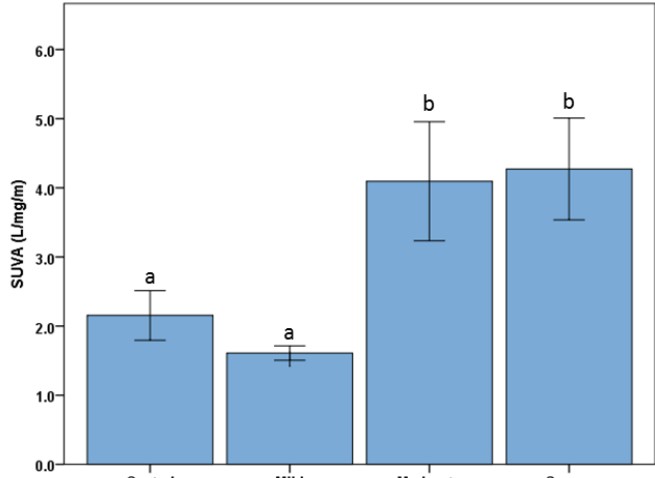


**Figure 3: SUVA value of *Molinia caerulea* derived DOC produced under differing severities of drought**

**(n=5 per treatment) with error bars at one standard error. Letters indicate statistically similar groups**

**from the Tukey test.**



**3.4 DOC removal efficiency**

Mean values for DOC removal by coagulation with ferric sulphate were in the order of *Juncus* ($54.7 \pm 2.3$ %),

*Molinia* ($37.5 \pm 2.6$ %), peat ($37.0 \pm 2.9$ %), *Calluna* ($35.1 \pm 2.0$ %) and then *Sphagnum* ($26.0 \pm 2.9$ %). The

Tukey HSD test suggested that the mean values for DOC removal efficiency fell into three subsets with similar

means in the order *Juncus> Molinia*, peat and *Calluna> Sphagnum*. The factorial ANOVA suggested no drought

effects on removal efficiency (p=0.418). The removal efficiency for all samples from each DOC source is

shown in Figure 4. *Juncus* DOC proved to be the easiest to remove via coagulation/flocculation with peat,

*Calluna* and *Molinia* all relatively easily removed at just under 40%. Comparatively poor removal was achieved

for *Sphagnum* DOC (<30%) which may be attributable to the low SUVA and peak C measure also found.







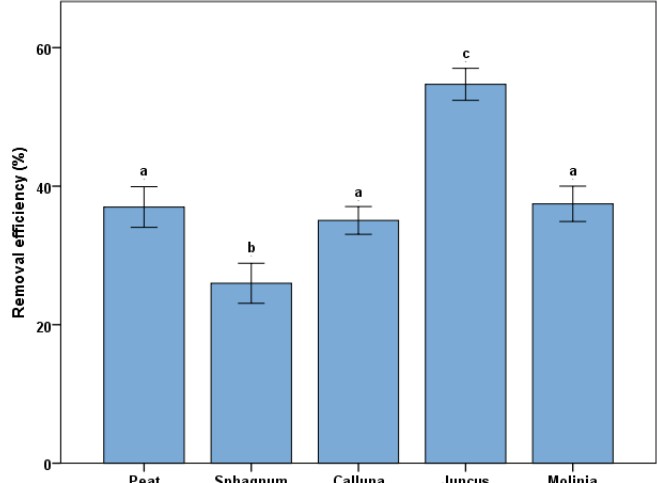


**Figure 4: DOC removal efficiency by coagulation/flocculation for different DOC sources (n=20 for each source, error bars at one standard error, letters indicate statistical subset according to Tukey test).**


### 3.5 SUVA removal efficiency

The removal of aromaticity, measured by SUVA, is of interest in drinking water treatment as aromatic compounds have a high propensity to form some of the regulated DBPs on chlorination (Bond et al., 2011). Large, aromatic compounds are selectively removed by coagulation/flocculation and as expected good removal (>70%) was observed for most of the samples. The mean values for the reduction in SUVA value following coagulation with ferric sulphate was in the order of peat (76.6 ± 1.8 %), *Sphagnum* (76.3 ± 2.5 %), *Molinia* (67.7 ± 4.7 %), *Calluna* (49.6 ± 5.3 %) and then *Juncus* (44.5 ± 2.3 %). The Tukey HSD test suggested that there were two subsets of DOC sources with similar means with peat, *Sphagnum* and *Molinia* > *Juncus* and *Calluna*. As with the overall DOC removal efficiency, there were no drought effects on SUVA removal (p=0.475). *Sphagnum* DOC showed good removal of SUVA despite relatively poor removal of total DOC, suggesting the aromatic compounds present in the sample are easily removed but that a large pool of aliphatic compounds are also present and these are more difficult to treat by conventional means.

319

### 3.6 Correlations between measures of DOC quality and treatability

A number of DOC quality indices based on absorbance and fluorescence measures were tested. The correlation coefficients for the different quality and treatability parameters are shown in Table 5. Peak C, a humic-like fluorescence peak, showed the best correlation with removal efficiency while the ratio of humic-like to protein-like fluorescence (Peak C/T) gave a lower but still significant correlation coefficient. The magnitude of peak C values were in the order *Juncus*>*Molinia*>*Calluna*>peat>*Sphagnum* which is consistent with data on removal efficiency. The SUVA value showed the best correlation with SUVA removal efficiency, suggesting that DOC with a lower proportion of aromatic compounds (low SUVA value) contains aromatic compounds which are



harder to remove by coagulation, possibly meaning they are either low molecular weight and/or also contain
hydrophilic groups.

**Table 5: Spearman's ρ for different DOC quality and treatability measures**

| DOC quality measure | Treatability measure | Spearman's ρ |
|---|---|---|
| **Peak C** | DOC removal % | 0.578, p<0.001 |
| **Peak C/T** | DOC removal % | 0.268, p=0.007 |
| **SUVA** | SUVA removal % | 0.445, p<0.001 |
| **Specific Peak C** | SUVA removal % | 0.235, p=0.019 |



**3.7 Repetition of control group**
The data obtained from DOC extracted before and after the repeated simulation were analysed using student's t-
test (equal variances assumed, confirmed using Levene's test) to assess whether the DOC extracted was
significantly different following six weeks of exposure to oxygen without any experimental treatment. The
results of this analysis are shown in Table 6.

**Table 6: t-tests for pre and post-incubation peat samples (significant differences highlighted in bold)**

| Variable | t test | p value | % change |
|---|---|---|---|
| **Extractable DOC** | **5.685** | **0.005** | **+41.6** |
| **Fluorescence peak C** | **8.168** | **0.011** | **-29.2** |
| **Fluorescence C/T** | 0.180 | 0.866 | Not significant |
| **SUVA** | **3.195** | **0.033** | **-23.0** |


Water extractable DOC increased significantly from 0.19 to 0.27 mg g$^{-1}$, an increase of 41.6%. The SUVA value
decreased at the end of the simulation from 3.62 to 2.85 L mg m$^{-1}$, as did the fluorescence Peak C measure,
which suggests a decrease in the level of aromaticity and humification of the DOC, respectively. This result may
explain why poorer DOC removal for peat DOC was observed in this experiment than in our previous work
(Ritson et al., 2016) as exposure to oxygen reduces the aromaticity of peat DOC and therefore it amenability to
removal via coagulation.

**4.0 Discussion**
**4.1 Water extractable DOC**
The peat soil was affected by the drought treatment with higher extractable DOC observed at the mild severity.
This finding is consistent with the 'enzymatic latch' hypothesis that increased oxygenation of peat engages a
biogeochemical cascade whereby increased phenol oxidase activity ends the phenol-induced inhibition of
hydrolase enzymes, thus increasing overall organic matter decomposition (Freeman et al., 2001a). This is also
confirmed by the replication of the control treatment which showed exposure to oxygen even in the absence of
drought increased DOC production and decreased DOC aromaticity. This finding has implications for all




laboratory studies which remove peat from anoxic conditions as these may not be representative of in-situ
conditions.
No effect was observed with the moderate and severe drought treatments which may be explained by water
scarcity limiting microbial activity (Toberman et al., 2008) and/or increased hydrophobic protection decreasing
the extractable DOC on rewetting. The very low final water content of the severe treatment and observations of
drying out and shrinkage of the peat mass throughout the experiment add weight to these possible explanations,
although actual rates of microbial respiration were not monitored during the experiment.
The lack of a drought effect on DOC production from any of the vegetation types suggest the pulse in DOC
observed post-drought elsewhere in catchment scale studies (Evans et al., 2005; Scott et al., 1998; Watts et al.,
2001; Worrall and Burt, 2004) is likely to be due to the oxygenation of peat soils rather than any litter layer
effects. This increase in peat-derived DOC is significant for downstream water treatment as our previous work
showed this has more environmental persistence than vegetation sources (Ritson et al., 2016) and the UV and
fluorescence data suggested DOC from peat exposed to oxygen may be more difficult to remove by
conventional treatment measures. High DOC production was noted for the vascular plants, suggesting they may
be an important source of DOC within peatland catchments during the period of their senescence, although
drought does not affect the amount they produce. Drought conditions may, however, precipitate a change in
vegetation type favouring more drought-tolerant species (Bragazza, 2008), which may have longer term effects
for peatland biogeochemistry.
The amount of DOC extracted from *Sphagnum* was low, which may be due to the fact that its litter is
recalcitrant to decay due to its high polyphenol content and numerous compounds with antimicrobial and
antifungal properties (van Breemen, 1995). The other typically upland species, *Calluna*, produced the second
least amount of DOC of the vegetation types, which also agrees with literature surrounding the recalcitrance of
its litter (Aerts, 1995; Huang et al., 1998) and field studies suggesting areas of *Calluna* produce more porewater
DOC than *Sphagnum* (Armstrong et al., 2012). The two grassland species, *Molinia* and *Juncus*, produced much
larger amounts of DOC per g of dry weight. This is in keeping with the growth strategy of these species,
whereby they rapidly produce a large amount of above-ground biomass and produce litter which decays readily,
providing a positive feedback to its strategy of rapid growth and fast nutrient cycling (Aerts, 1999; Mann and
Wetzel, 2000). This growth strategy is in contrast to that of the upland species *Calluna* and *Sphagnum*, which
have adapted to low nutrient availability and therefore grow slowly, have nutrient poor litter and invest fewer
resources in material which cycles rapidly (Aerts, 1999). Correlations between litter C:N ratio, suggesting
nutrient availability, and amount of extractable DOC have been found in our previous work (Ritson et al., 2016)
and elsewhere in the literature (Soong et al, 2014).
*Molinia* encroachment is a well acknowledged problem in Europe (Chambers et al., 2007b; Heil and Diemont,
1983; Hughes et al., 2007; Milligan et al., 2004) and nitrogen deposition and drier summers may mean more
grassland species in the UK uplands in the future. The results of this study suggest the transition from
*Sphagnum* to *Calluna* and *Molinia* observed in a paleoecological study of the area nearby our Exmoor site
(Chambers, 1999) may have increased the amount of extractable DOC in the litter layer on g per g basis, as well
as increased the seasonality of its export (Ritson et al., 2016). The much greater effect sizes for DOC source
versus drought controls in this study and temperature and rainfall controls in previous work (Ritson et al.,
2014a) suggest that the source of the DOC may be the primary driver of DOC quantity and quality in peatland





litters, consistent with litter decomposition studies in boreal peatlands (Straková et al., 2011). This has important
implications for overall soil carbon stability in peatlands as the addition of labile carbon from litter can stimulate
the decomposition of older carbon (Fontaine et al., 2007).
Studies concerning vegetation control of pore-water DOC are limited, but are reviewed in Ritson et al. (2016).
Fenner et al. (2007) found elevated $CO_2$ caused a transition from *Sphagnum* to *Juncus* dominance on monoliths
from flush peat which gave a 66% rise in DOC, attributed to an increase in above-ground biomass, more labile
litter and stimulation of peat decomposition through root exudation. Vestgarden et al., (2010) found DOC in
pore-waters beneath different vegetation types to be in the order *Molinia>Calluna>Sphagnum* in shallow
samples but *Sphagnu*m had higher concentrations than the vascular plants at depth and showed less seasonal
variation. This has been linked to the seasonal growth cycles of vascular plants in peatlands which provide litter
which decomposes rapidly and produces a large amount of DOC on a mg per g basis creating greater seasonality
in DOC export (Ritson et al., 2016).
**4.2 SUVA**
The SUVA value has been linked to the aromaticity of DOC (Weishaar et al., 2003) and is of interest as a
predictor of coagulation removal efficiency and DBP formation (Matilainen et al., 2011) in water treatment. The
highest SUVA value was observed for the peat soil and *Molinia* litter, and the lowest value for the statistical
subset of *Sphagnum* and *Calluna*. In a similar trend to DOC production, it appears that the grassland species
produce DOC of greater aromaticity than the peatland species. *Molinia* also showed an interactive effect with
the drought treatment, with a greater flux of aromatic compounds at the moderate and severe treatments,
suggesting dry conditions are favourable for the breakdown and/or solubilisation of aromatic compounds in
*Molinia* litter. *Molinia* DOC may, therefore, contribute to the increase in the aromaticity of peatland DOC
observed after droughts at the catchment scale (Scott et al., 1998; Watts et al., 2001), although solubility
controls on peat-derived DOC may be more important (Clark et al., 2006, 2005; Clark et al., 2011).
No drought effect was found for the SUVA value of peat which is in contrast to field studies which have shown
a decrease in aromaticity of DOC during drought due to solubility controls and an increase in aromaticity on
rewetting (Evans et al., 2005; Scott et al., 1998; Watts et al., 2001; Worrall et al., 2004). This may be explained
by the fact that field studies have shown an increase in DOC aromaticity over many years, whereas this study
examined a single rewetting event following drought, so the altered biogeochemical controls on DOC
aromaticity may not have had enough time to exert a significant effect. The laboratory conditions may also have
played a part, as the control sample is likely to have been exposed to more oxygenation through sample
collection and setup of the experiment than undisturbed peat in the field, therefore increasing its similarity to the
treatment conditions. The changes in DOC properties when the control group was repeated would appear to
confirm this hypothesis.
These results suggest encroachment of grassland species into the uplands will increase seasonal DOC flux from
the litter layer and increase the aromaticity of exported DOC and create a drought effect where *Molinia* litter is
present. The lack of a drought effect for peat suggests that the long-term effects caused by water table
drawdown identified elsewhere in the literature will likely be more important for DOC flux than the short-term
effects studied here.





### 4.3 DOC and SUVA removal

DOC removal for all sources were typical of literature values (Matilainen et al., 2010), with *Juncus* DOC proving the easiest to remove and *Sphagnum* DOC the hardest. Repeating the control condition and measuring DOC production and quality parameters allowed an estimate of the effect of oxygen exposure for peat samples. This showed a decrease in SUVA value and humic-like character (fluorescence Peak C) as well as a large increase in extractable DOC. These changes in quality parameters may provide an explanation of why poorer removal by coagulation was achieved for peat following this drought experiment than had been observed in our previous work (Ritson et al., 2016) as less aromatic/humified material is likely to be harder to remove by coagulation (Bond et al. 2011). Poorer removal was observed for *Sphagnum* than in our previous work; the effect of more oxygenated conditions on vegetation decomposition remains an area for further research, particularly as climate change may increase the likelihood of water table draw down in peatlands. The coagulation removal efficiency could best be explained by the Peak C fluorescence index, suggesting humic substances content was the strongest predictor of DOC removal. This is in contrast to our previous work which found the ratio of humic to protein-like DOC to be the most important predictor (Ritson et al. 2014b). Our previous work used DOC collected throughout a two-month simulation rather than a single re-wetting event at the end. The samples will, therefore, have likely undergone microbial processing during this simulation and consequently an increase in the amount of autochthonous DOC, hence the greater importance of the fluorescence measure of protein-like DOC.

### 5.0 Conclusions

Climate projections for the UK vary, however most agree the likelihood of droughts in the future is set to increase. The results of this research suggest the dominant effect of drought on peatland DOC sources is to increase the amount and decrease the treatability of DOC from peat soils. This is likely due to the 'enzymatic latch' mechanism increasing decomposition when oxic conditions prevail. No drought effect on different vegetation litters was found, suggesting that the greatest effect of drought for vegetation may be facilitating shifts to drought-tolerant species dominance rather than altering decomposition processes in the short term. Oxygenation of peat appears to greatly increase extractable DOC whilst also decreasing the aromaticity and humification, which may mean it is more difficult to remove at the treatment works. These results provide support for catchment management programmes seeking to increase resilience to drought by raising peatland water tables as a strategy for mitigating against high riverine DOC concentrations following droughts.

### Author contributions

All authors developed the experimental design and advised on the subsequent analysis. Ritson performed the experiments and data analysis. The manuscript was written by Ritson with contributions from all co-authors.

### Acknowledgements

This work was supported by the Engineering and Physical Sciences Research Council [grant number EP/N010124/1]. The authors would also like to thank the Grantham Institute: Climate and Environment and Climate-KIC for the financial support of Jonathan Ritson. The authors would also like to thank South West





Water's Mires project for access to sites as well as Exmoor and Dartmoor National Park Authorities, Natural
England and Duchy of Cornwall. Freeman acknowledges NERC Grant NE/K01093X/1.

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
