# Peer review of "The effect of drought on dissolved organic carbon (DOC)"

_Biogeosciences, 2016_

## Referee Comment (RC1) · Anonymous Referee #1 · 23 Jan 2017

The paper reports amounts of DOC extracted from peat and from 4 different plant species (5 sources) after these samples were subjected to different irrigation intensities followed by water extraction. In addition SUVA spectra of the extracts were determined and the coagulation potential of DOC was measured. The aim was to investigate the effects of drought on DOC release and properties. The authors draw far reaching conclusions on the effects of drought and vegetation change on peatlands DOC budgets. I am not in favor of publication of this study because of methodical shortcomings, the small data base, inconsistent presentation of data, and over interpretation of results. 1 The study is based on the analysis of only 100 water samples for easily measureable parameters. Hence the data base is very small and the interpretation of the patterns

of DOC release and DOC quality suffers from the lack of measuring any explanatory variable like e.g. CO2 release, data on microbial or enzymatic activities in the different treatments, chemistry of the plant material used, pH or other relevant chemical parameters of extracts, etc. Overall, the paper remains highly descriptive and - for an international audience - provides too little innovation on DOC dynamics in peatlands. 2 Moreover, the experimental approach is strange: Samples were subjected in the lab to irrigation at different rates for 6 weeks to induce different drying intensities. The degree of desiccation after and during the 6 weeks was not measured, nor the biological status of the samples. Only for the peat samples some data on water contents at the end of irrigation (unit?) are given in line 251. 3 The different intensity of irrigation should induce different leaching rates and different DOC fluxes from the samples. No information is given on that. 4 Following the 6 weeks of irrigation, all samples were air dried before water extraction (line 148) which does not make sense to me: If all samples were air dried before extraction, the pre irrigation to induce different degrees of desiccation seems meaningless. The rewetting of air dried soil samples cause specific effects (Birch effects) that my override the aimed irrigation effect. 5 The data presentation needs substantial revision: The content of tables 1 – 6 and the main message can easily be given in text form (tables 1-6 can be omitted). Fig. 1 gives DOC release from the 5 sources, Fig. 2 gives drought effects on only peat samples, Fig 3 gives SUVA only for Molinia, Fig 4 gives removal efficiency for the 5 sources, but without drought effects. Hence, the presentation is confusing and inconsistent. 6 The conclusions on effects of climate and vegetation change on peatland biogeochemistry are highly speculative in view of this short term laboratory study.

---

## Referee Comment (RC2) · Anonymous Referee #2 · 27 Jan 2017

**General comments**

The manuscript «The effect of drought on dissolved organic carbon (DOC) release from peatland soil and vegetation sources» addresses relevant scientific questions related to water quality, climate change, drinking water and catchment management. The work builds onto previous work by these authors and others, but adds new aspects and brings the understanding forward. The experimental setup is adequate, although extractions also before treatment would have been beneficial (hence the add-on experiment would not have been needed). The method is usually clearly explained, and the manuscript is well structured. The figures and tables are adequate, and although the number of samples varies between figures, this is clearly explained. The data analysis is generally sound. The major objections are related to the limited explanation of the how the add-on experiment relates to the main experiment, including how the add-on experiment can explain the different results for control samples in this experiment and the previous Ritson et al. (2016) - and part of the discussion of the drought effect for peat soils, including the wider consequences of the findings. However, I find these kinds of controlled experiments valuable, and after considering carefully the points below I regard it as worth publishing.

**Specific comments**

A drought effect is observed in peat soil, but the interpretation could be elaborated/adjusted:

- The abstract states in line 29-30 that "more immediate effects are observed in peat soils". This is correct, but if drought events will be more frequently observed in the future, these pulses of DOC can also be regarded as a long-term effect, in that they will be occurring more frequently, potentially giving a steady increase in DOC concentration.
- It is somewhat surprising that drought effect was only observed with the mild treatment. This is explained by large variability in the other treatments, possibly because some samples became drier than intended (line 244-261). The arguments are mainly repeated in lines 359-363, but I miss a discussion of the implications of this. Do these results indicate that there is an "optimum" drought frequency for DOC release, i.e. that DOC release will not increase with increasing drought frequency and severity, but will increase to a certain point and then decline?
- Line 423-426: Are you suggesting that drought causes permanently altered biogeochemical controls so that the released DOM becomes gradually more aromatic? The literature usually argues that more aromatic DOM is released after single drought events, but that increased frequency of these will give increased aromaticity over time. Please explain in more detail in which way you suggest your single rewetting differs from field studies and how this may have affected the results.
- In line 431-435 the results on both DOC and SUVA seem to be summarized. Do you consider that there was a "lack of drought effect for peat" or are you here only talking about SUVA? And again, you argue that the experiment simply investigates short-term effects. It is true, in the sense that only one single drought event is mimicked. But are there arguments that long-term effects of drought go beyond the sum of many single events, that there are more permanent changes going on? This is what you indicate, but you do not explain or express it clearly.

The conditions of the control group were repeated, but this is not clearly justified/explained, and the interpretation can be questioned

- The title of section 2.5 should rather be "Repetition of the control group conditions"

- Line 186-192: Please explain why peat samples for this additional test were collected at a different site. And explain more clearly why this extra experiment was performed? Was it simply because in the main experiment there was no extraction prior to treatment, so you did this to look at changes over the course of the experiment?
- Line 337: I would change "without any experimental treatment" to "at control conditions" (which are certainly experimental in some sense)
- Line 344-347: Delete this type of discussion text from the results chapter
- Line 439-447: The discussion comes here, but it is not clear. Yes, you show that DOC removal may decline with time due to change in DOM properties, but it is not clear why this suggests that DOC removal was lower in this experiment than in Ritson et al. (2016). As far as I can see the control samples in the current experiment underwent exactly the same treatment as the peat samples in the previous experiment. Figure 4 shows DOC removal across treatments, but the results for the control group given in the supplement should be directly comparable to Figure 1 in the 2016 paper – which shows a big difference in DOC removal. I cannot see that this follow-up experiment explains why there is such a big difference. This is important, as you argue (e.g. in the abstract) that DOC from peat is harder to remove, but in Ritson et al. (2016) it is the easiest to remove. Please elaborate
- Line 463-464: It is claimed that drought (oxygenation) deceases aromaticity, while the drought experiment itself did not show effects on DOM quality for peat soils. You argue why this may be so in section 4.2, but please repeat it briefly here and modify the conclusions (make them less firm).

Line 76: You may mention what kind of programmes/how Sphagnum dominance is promoted

Line 139: I assume the intervals between the rainfall simulation were the same, but please specify this

Line 152: I assume the extracts were filtered before further analysis? Please explain

Line 167: Was the coagulation performed on filtered samples? In case, please justify this

Line 207-208: Move and merge into 3.1 to avoid repeating this information here

Line 219-222: Move the more detailed explanation of the method to section 2.4

Line 230-231: Specify that you are talking about the peat soil

Line 276-278: Specify that you are talking about the Molinia samples

Section 3.4: Move the sentence in line 296-297 (on drought effects) to the beginning of the section. Line 299-300 simply repeats line 293-294 – move and merge.

Line 308-310: Move to the discussion (section 4.2)

Section 3.6: The fluorescence data are only presented in connection with coagulation. But what about difference in fluorescence properties related to drought treatment or vegetation type? Why are these results not presented and discussed?

Line 367-369: This probably relates to the results given in table 6, but it does not fit with the lack of drought effect on peat SUVA. I suggest just briefly mentioning this here, but refer to the lack of drought effect for peat discussed in section 4.2

Line 460-462: You could mention the drought effect on SUVA for Molinia, which may partly counteract the oxygenation effect of peat (lower aromaticity)

**Technical corrections**

Line 139: Space between "applied" and "eleven" missing.

Line 151-2: "Kalbitz" incorrectly spelled, and both references missing in the reference list. And why is the reference not put at the end of the sentence? Is the latter part the author's own interepretation?

Line 155: Unit misspelled, should be mgC $l^{-1}$.

Line 266: Delete "group two Calluna and"

Line 278: Replace "medium" with "moderate", as this is the term used elsewhere

Line 323: Add DOC before "removal efficiency"

Line 344+line 346 and similar places: When talking about the properties of the actual molecules in question, use DOM, not DOC. DOC is just a notation for what is actually analysed and for which we can talk about changes in concentration etc, but DOC cannot be more or less aromatic or humified.

Line 433: Add "SUVA" before peat

References: Sometimes access date is added, sometimes not. In general web page and access date should not be necessary for published papers, but at least be consistent.

---

## Author Comment (AC1) · 27 Jan 2017

We thank the reviewer for their comments on our manuscript and will attempt to address these issues. Below is a detailed response to each of the six issues the reviewer has highlighted.

Point 1: 'The study is based on the analysis of only 100 water samples for easily measureable parameters'

Response: Although it would be desirable to include as many samples as possible in the experimental design this was limited to 5 replicates per vegetation/treatment for practical considerations. Similar studies looking at decomposition have used similar,

or indeed lower, numbers of replicates. Soong et al. (2015) used three replicates per substrate in a laboratory decomposition experiment concerning DOC from fresh and pyrolysed litter. Fellman et al. (2015) also used three replicates per substrate in a litterbag study of litter decomposition whilst Cleveland et al. (2004) performed six DOC extractions per litter type and then bulked them to create three replicates. In a laboratory study on the decomposition of Calluna Vulgaris, Van Meeteren et al. (2007) used five replicates per treatment in a similar approach to our own study.

We feel five replicates, giving 100 samples in total, is a good balance between capturing natural variability in the samples and practicality given the samples filled three climate control cabinets and that manual irrigation of the samples (to ensure even wetting of the vegetation/soil) was necessary. In the subsequent analysis 200 samples were analysed for TOC and UV properties (pre- and post-coagulation) as well as 100 fluorescence samples. The coagulation experiments themselves were time consuming as commercial jar-testing apparatus is limited to six samples per run and each run takes ∼2hrs to perform. Although we were unable to present the data due to quality concerns resulting from instrument failure, 100 samples with two replicates (200 total) were also chlorinated, quenched and then extracted to assess disinfection by-product formation.

The reviewer notes the lack of any supporting water chemistry or litter chemistry data in the paper. We apologise for this oversight as we did measure pH of the extracts and will include these data in an updated version of the manuscript. We also measured the carbon and nitrogen content of a sub-sample of the starting soil/litter, however we did not include this in the manuscript as correlations between C:N and extractable DOC were shown in out 2016 paper in Scientific Reports. We instead referred to this paper in the discussion section. We will include the C:N data in an updated version of the manuscript. Although measuring $CO_2$ production during the experiment would have been desirable we already acknowledge the lack of these measurements as a weakness in our ability to confirm the cause of changes in extractable DOC between

drought treatments (line 363).

Point 2: The degree of desiccation after and during the 6 weeks was not measured, nor the biological status of the samples. Only for the peat samples some data on water contents at the end of irrigation (unit?) are given in line 251.

Response: The unit of the final water content is grams and is given in the text. Data on final water content for the peat soil and Sphagnum litter are available and confirm the efficacy of the irrigation treatment in causing degrees of desiccation in the treatments. This can be included in the updated manuscript.

Point 3: The different intensity of irrigation should induce different leaching rates and different DOC fluxes from the samples. No information is given on that.

Response: Unfortunately, DOC from each irrigation event was not recorded. We note in the method section that:

'Previous work has shown that the amount of water used to extract DOC and whether one extraction is performed or sequential extractions to simulate multiple rainfall events gives no significant variation in DOC quality (Don and Kalbtiz, 2005, Soong et al., 2014), only changes in the total amount of carbon'.

We would therefore suggest that any differences in DOC quality are captured by our approach. Rewetting following drought of interest as this has been highlighted in the literature (and discussed in our introduction) as a period of increased riverine DOC concentrations. One of the goals of the experiments was to ascertain whether litter layer DOC flux played a role in the increased DOC concentrations post-drought or whether this was entirely due to processes within the peat, hence our focus on rewetting.

Point 4: Following the 6 weeks of irrigation, all samples were air dried before water extraction (line 148) which does not make sense to me: If all samples were air dried before extraction, the pre irrigation to induce different degrees of desiccation seems meaningless. The rewetting of air dried soil samples cause specific effects (Birch effects) that my override the aimed irrigation effect.

Response: Air-drying was performed so that accurate estimates of extractable DOC could be determined on a mgC g-1 basis. Whilst air drying the samples after the irrigation simulations may have increased the homogeneity between the sample treatments, we feel this is likely to be minor as this occurred for approximately 2-3 days compared to a 42 day simulation. Also, during the simulation all samples will have been exposed to periods with no irrigation multiple times due to the number of days of rainfall being fixed across all treatments. The differences between treatments, therefore, are the extent of decomposition and DOC production during the 42 day simulation due to desiccation rather than the final water content.

Point 5: a) The data presentation needs substantial revision: The content of tables 1 – 6 and the main message can easily be given in text form (tables 1-6 can be omitted).

Response: We can reduce the number of tables in the manuscript if the editor feels this is necessary. Table 1 can be described easily in the text so may be removed, however Table 2 contains twelve p values as well as eight $\omega 2$ values so we feel a table is appropriate to summarise this information for the reader. It would also be possible to incorporate Tables 5 and 6 into the text if necessary.

Point 5: b) Fig. 1 gives DOC release from the 5 sources, Fig. 2 gives drought effects on only peat samples, Fig 3 gives SUVA only for Molinia, Fig 4 gives removal efficiency for the 5 sources, but without drought effects. Hence, the presentation is confusing and inconsistent.

Response: The reasoning for this is explained in the text as treatment and/or interactive effects are interrogated. In Fig 2 drought effects were shown only for peat as this was the only DOC source with significant drought effects. Similarly, only Molinia was included in Fig 3 as this was the only source with a significant drought effect on SUVA. Finally, drought effects were not included in Fig 4 as there were no significant drought effects for removal efficiency.

Point 6: The conclusions on effects of climate and vegetation change on peatland biogeochemistry are highly speculative in view of this short term laboratory study.

Response: Below is a sentence by sentence justification of our conclusions section.

"Climate projections for the UK vary, however most agree the likelihood of droughts in the future is set to increase."

This point is not from our data yet is supported by research cited in the introduction.

"The results of this research suggest the dominant effect of drought on peatland DOC sources is to increase the amount and decrease the treatability of DOC from peat soils" This is supported by our data. The drought treatment only effected the peat soils, increasing the amount of extractable DOC. The repetition of the control group showed that exposure to oxygenation at any level decreases the SUVA value and humification (measured by peak C) which are both commonly used as indicators of the treatability of DOC.

"This is likely due to the 'enzymatic latch' mechanism increasing decomposition when oxic conditions prevail" Although we present this as a likely hypothesis we have acknowledged in the earlier text that as CO2 production was not measured this cannot be confirmed, hence use of the word 'likely'.

"No drought effect on different vegetation litters was found, suggesting that the greatest effect of drought for vegetation may be facilitating shifts to drought-tolerant species dominance rather than altering decomposition processes in the short term."

Here we note that no effects were found for drought on DOC production from vegetation, whilst qualifying that this was a short term study. We fit our findings into the wider literature which suggests drought may facilitate shifts in species dominance in peatlands and there may be, therefore, longer term effects from droughts. Comparison between the typical peatland species (Sphagnum and Calluna) and the more drought tolerant ones (Molinia and Juncus) in our study sheds some light on these longer term

effects.

"Oxygenation of peat appears to greatly increase extractable DOC whilst also decreasing the aromaticity and humification, which may mean it is more difficult to remove at the treatment works." Here we restate our primary finding which is supported by the data we present.

"These results provide support for catchment management programmes seeking to increase resilience to drought by raising peatland water tables as a strategy for mitigating against high riverine DOC concentrations following droughts."

In this final statement we extrapolate our findings into the context of peatland restoration occurring in the UK and elsewhere. As our primary finding is that oxygenation of peat leads to more DOC which is harder to treat, we suggest that management that seeks to limit the extent and/or frequency these conditions occur is positive from a water treatment point of view. We do not make any other conclusions regarding these schemes.

References Cleveland, C.C., Neff, J.C., Townsend, A.R., Hood, E., 2004. Composition, Dynamics, and Fate of Leached Dissolved Organic Matter in Terrestrial Ecosystems: Results from a Decomposition Experiment. Ecosystems 7, 275–285. doi:10.1007/s10021-003-0236-7

Fellman, J.B., Petrone, K.C., Grierson, P.F., 2013. Leaf litter age, chemical quality, and photodegradation control the fate of leachate dissolved organic matter in a dryland river. J. Arid Environ. 89, 30–37. doi:10.1016/j.jaridenv.2012.10.011

Soong, J.L., Parton, W.J., Calderon, F., Campbell, E.E., Cotrufo, M.F., 2015. A new conceptual model on the fate and controls of fresh and pyrolized plant litter decomposition. Biogeochemistry. doi:10.1007/s10533-015-0079-2

Van Meeteren, M., Tietema, A., Westerveld, J., 2007. Regulation of microbial carbon, nitrogen, and phosphorus transformations by temperature and moisture during decomposition of Calluna vulgaris litter. Biol. Fertil. Soils 44, 103–112. doi:10.1007/s00374-007-0184-z

---

## Referee Comment (RC3) · Anonymous Referee #3 · 31 Jan 2017

General Comments

The authors address relevant questions regarding DOC leaching/quality from peat and various vegetation following different severities of drought and the potential impact this DOC may have on water quality. Implications from this study suggest that DOC quantities will increase under drought conditions while its treatability in water management will decrease. Authors advocate for catchment management to minimize drought effects on peatlands in light of the results presented here. Experimental design and replication seems adequate to address their research questions. Methods are well structured and fairly easy to follow. Tables and Figures are adequate and useful to the reader. However, authors need to be more conscious of separating their Discussion

from their Results. Nearly every Results sub-section finished with an observation of why they got such results. These observations are the reason manuscripts have a Discussion section. Report results only in the Results section and discuss why you may have gotten those results and what their implications may be in the Discussion (see specific comments). Also, I don't understand why the authors included a repetition of the control group. I don't think this side-experiment is necessary and doesn't add value to the main experiment except as validation. $CO_2$ measurements would also be beneficial to validate their reliance on the enzymic-latch hypothesis, without which it is only an assumption. Significant improvements could be made in grammar and phrasing throughout the manuscript. Overall, the methods justify the results and the results seem to justify most of the discussion and implications drawn from this study. The authors do tend to over-interpret the magnitude of their results on future drought effects given this short-term laboratory study, but I still find merit in this study and recommend publication following revisions on the comments listed below.

Specific Comments

Lines 38-41: Vague sentences that aren't useful to a reader as written.

Line 42: "represents a significant flux of carbon from peatlands (Dinsmore et al. 2010)" - Provide a range of flux values rather than another vague sentence. Don't make the reader dig into every one of your citations to find useful information that could have easily been supplied.

Line 76-77: Vague sentence. Provide useful information from this citation that explains which programmes are being promoted to increase Sphagnum dominance.

Line 219-222: Description of Holm-Sidak correction should be moved to Methods section "2.4 Data analysis and statistical methods"

Line 234-236: Not a result. Move to Discussion.

Line 245-249: Not a result. Move to Discussion.

Line 253-254: Not a result. Move to Discussion.

Line 258-261: Not a result. Move to Discussion.

Line 281-283: Not a result. Move to Discussion.

Line 316-318: Not a result. Move to Discussion.

Line 335-337: Do not introduce a new statistical test in the Results. Move this entire description to the Methods section "2.4 Data analysis and statistical methods"

Technical Corrections

Inconsistent use of "carbon" and "C" throughout the manuscript. Write "carbon (C)" the first time it is used and "C" afterwards.

Line 59: Change "effects" to "affects"

---

## Author Comment (AC2) · 27 Feb 2017

We thank the reviewer for their comments on our manuscript and will attempt to address these issues. Below are the main points raised by the reviewer and our responses to them. The reviewer also made a number of minor comments and technical points which we do not address here but will be happy to incorporate into the updated manuscript.

Point 1: The abstract states in line 29-30 that "more immediate effects are observed in peat soils". This is correct, but if drought events will be more frequently observed in the future, these pulses of DOC can also be regarded as a long‐term effect, in that they will be occurring more frequently, potentially giving a steady increase in DOC concentration.

Response: We agree with the reviewer's comment and will amend the manuscript to highlight this.

Point 2: It is somewhat surprising that drought effect was only observed with the mild treatment. This is explained by large variability in the other treatments, possibly because some samples became drier than intended (line 244-261). The arguments are mainly repeated in lines 359-363, but I miss a discussion of the implications of this. Do these results indicate that there is an "optimum" drought frequency for DOC release, i.e. that DOC release will not increase with increasing drought frequency and severity, but will increase to a certain point and then decline?

Response: Yes, we hypothesise that this is due to 'water scarcity limiting microbial activity (Toberman et al., 2008) and/or increased hydrophobic protection decreasing the extractable DOC on rewetting'. We would suggest that at very severe levels of drought DOC production is limited by water scarcity, however this would not stop oxygenation of peat and therefore greater potential for increased DOC production in the future due to the enzymatic latch mechanism. We will add a more detailed explanation of the implication of this finding to the amended manuscript.

Point 3: Line 423-426: Are you suggesting that drought causes permanently altered biogeochemical controls so that the released DOM becomes gradually more aromatic? The literature usually argues that more aromatic DOM is released after single drought events, but that increased frequency of these will give increased aromaticity over time. Please explain in more detail in which way you suggest your single rewetting differs from field studies and how this may have affected the results.

Response: In this section we discuss the lack of an increase in SUVA value for peat from the drought simulation, in contrast to field studies. The literature often suggests that DOC is elevated for many years after drought events. As we were monitoring a single rewetting event we suggest that one of the possible explanations for conflicting results could be that many of the longer term processes involved in increased DOC

concentration and aromaticity (enzymatic latch, recovery from sulphate acidification from oxygenation) may not have had time to occur. We will explain this more clearly in the updated manuscript.

Point 4:In line 431-435 the results on both DOC and SUVA seem to be summarized. Do you consider that there was a "lack of drought effect for peat" or are you here only talking about SUVA? And again, you argue that the experiment simply investigates short‐term effects. It is true, in the sense that only one single drought event is mimicked. But are there arguments that long‐term effects of drought go beyond the sum of many single events, that there are more permanent changes going on? This is what you indicate, but you do not explain or express it clearly.

Response: Yes, as this is in the section headed 'SUVA' we were only referring to effects on SUVA in this statement. We will clarify this in the updated manuscript. The reviewer is correct that our intention was to suggest that frequent droughts could create long term changes in peatland biogeochemistry, but that our experiment did not cover this. Again, we would be happy to clarify this in the updated manuscript.

Point 5: Line 186-192: Please explain why peat samples for this additional test were collected at a different site. And explain more clearly why this extra experiment was performed? Was it simply because in the main experiment there was no extraction prior to treatment, so you did this to look at changes over the course of the experiment?

Response: The reasoning behind performing the extra experiment was to interrogate the possibility that any oxygenation of peat could affect DOC quantity and quality and thus explain differences between the results found here and our previous work. The reviewer is correct that this could have been achieved by extracting DOC from a sub-sample prior to the start of the original experiment, however as this was not done we performed this short experiment. The samples were collected from an ombrotrophic peatland with a comparable mixture of vegetation (Juncus, Molinia, Sphagnum, Calluna, Eriophorum) and were of the same level of humification (von Post scale). Although not identical to the peat collected in the original experiment, we feel these samples are similar enough to test the hypothesis that the control conditions used in the original experiment give enough oxygenation to alter DOC properties.

Point 6: Line 439-447: The discussion comes here, but it is not clear. Yes, you show that DOC removal may decline with time due to change in DOM properties, but it is not clear why this suggests that DOC removal was lower in this experiment than in Ritson et al. (2016). As far as I can see the control samples in the current experiment underwent exactly the same treatment as the peat samples in the previous experiment. Figure 4 shows DOC removal across treatments, but the results for the control group given in the supplement should be directly comparable to Figure 1 in the 2016 paper – which shows a big difference in DOC removal. I cannot see that this follow‐up experiment explains why there is such a big difference. This is important, as you argue (e.g. in the abstract) that DOC from peat is harder to remove, but in Ritson et al. (2016) it is the easiest to remove. Please elaborate.

Response: The confusion here lies in the experiment we are referring to in Ritson et al. (2016) as there are multiple experiments in this paper. The control group of this paper is directly comparable to the experiment entitled 'Litter decomposition in the laboratory' in the Ritson et al. 2016 paper where only data on amount of DOC extracted were presented. The comparison we were intending to make, however, is to the first experiment from the 2016 paper entitled 'Ease of DOC removal during the treatment process for different peatland sources'.

In the 2016 coagulation experiments DOC was extracted from fresh peat which had had minimal exposure to oxygen. We suggest a reason why the peat DOC in the 2017 paper showed poorer removal by coagulation was that it had been exposed to oxygen over the length of the simulation and this may have altered the treatability of the extracted DOC. The repetition of the control group conditions provides evidence for this as it shows exposure to oxygen causes a decrease in Peak C and SUVA, both of which have been correlated with ease of removal via coagulation in the literature. We

will explain this in greater detail in an updated version of the manuscript and make it clear that when we say in the abstract that that peat DOC is harder to remove we mean peat that has been exposed to oxygen compared to peat which has not.

Point 7: Section 3.6: The fluorescence data are only presented in connection with coagulation. But what about difference in fluorescence properties related to drought treatment or vegetation type? Why are these results not presented and discussed?

Response: These data are available and can be included in the updated manuscript. The data suggest a drought effect on Peak C (humic-like) fluorescence for both Molinia and Juncus and an effect on Peak T (protein-like) fluorescence for Juncus.

Point 8: Line 367-369: This probably relates to the results given in table 6, but it does not fit with the lack of drought effect on peat SUVA. I suggest just briefly mentioning this here, but refer to the lack of drought effect for peat discussed in section 4.2

Response: The statistical subset of peat and Molinia had the highest SUVA values and we link this to our previous work suggesting this is likely to mean higher environmental persistence. We will rephrase this to make this clear and mention the lack of a drought effect on peat SUVA.

Point 9: Line 460-462: You could mention the drought effect on SUVA for Molinia, which may partly counteract the oxygenation effect of peat (lower aromaticity).

Response: Agreed.

---

## Author Comment (AC3) · 27 Feb 2017

We would like to thank the reviewer for taking the time to comment on our manuscript. The reviewer's main points stem from the inclusion of some discussion in the results section and our reasoning for repeating the control conditions.

We will be happy to separate the results and discussions more clearly in an updated manuscript and also address the 'specific' and 'technical' comments listed by the reviewer. The reasoning for performing our additional experiment has been covered in our response to reviewer #2. As we found differences in removal efficiency via coagulation between this experiment and our previous work (Ritson et al. 2016), we wanted to test the hypothesis that any amount of oxygenation can affect DOC quality from

peat soils. We feel removing this experiment from the manuscript would detract from its value as we would then be only speculating on the reasons for the difference in findings between the two papers.

---

## Author Response (AR1)

[revised manuscript text omitted]

**Response to reviewer comments and changes made**

Reviewer #1

**Point 1: 'The study is based on the analysis of only 100 water samples for easily measureable parameters'**

Our response in the online discussion: "Although it would be desirable to include as many samples as possible in the experimental design this was limited to 5 replicates per vegetation/treatment for practical considerations. Similar studies looking at decomposition have used similar, or indeed lower, numbers of replicates. Soong et al. (2015) used three replicates per substrate in a laboratory decomposition experiment concerning DOC from fresh and pyrolysed litter. Fellman et al. (2015) also used three replicates per substrate in a litterbag study of litter decomposition whilst Cleveland et al. (2004) performed six DOC extractions per litter type and then bulked them to create three replicates. In a laboratory study on the decomposition of *Calluna Vulgaris*, Van Meeteren et al. (2007) used five replicates per treatment in a similar approach to our own study.

We feel five replicates, giving 100 samples in total, is a good balance between capturing natural variability in the samples and practicality given the samples filled three climate control cabinets and that manual irrigation of the samples (to ensure even wetting of the vegetation/soil) was necessary.
In the subsequent analysis 200 samples were analysed for TOC and UV properties (pre- and post-coagulation) as well as 100 fluorescence samples. The coagulation experiments themselves were time consuming as commercial jar-testing apparatus is limited to six samples per run and each run takes ~2hrs to perform. Although we were unable to present the data due to quality concerns resulting from instrument failure, 100 samples with two replicates (200 total) were also chlorinated, quenched and then extracted to assess disinfection by-product formation.

The reviewer notes the lack of any supporting water chemistry or litter chemistry data in the paper. We apologise for this oversight as we did measure pH of the extracts and will include these data in an updated version of the manuscript. We also measured the carbon and nitrogen content of a sub-sample of the starting soil/litter, however we did not include this in the manuscript as correlations between C:N and extractable DOC were shown in out 2016 paper in *Scientific Reports*. We instead referred to this paper in the discussion section. We will include the C:N data in an updated version of the manuscript. Although measuring $CO_2$ production during the experiment would have been desirable we already acknowledge the lack of these measurements as a weakness in our ability to confirm the cause of changes in extractable DOC between drought treatments (line 363)."

Corrections made:

Added to section 2.2: 'Similar experiments concerning the decomposition of litter have used three replicates per treatment (Fellman et al., 2013; Soong et al., 2015), suggesting our approach of using five samples per treatment is adequate to capture variability between samples.'

pH data added to Table 2 and section 3.2

Section 2.3 added: 'Elemental analysis on a subsample of the starting material revealed C:N to be in the order peat (29.9), *Molinia* (35.7), *Juncus* (42.2), *Calluna* (56.5) and *Sphagnum* (93.7) as reported in Ritson et al. (2016).'

**Point 2: The degree of desiccation after and during the 6 weeks was not measured, nor the biological status of the samples. Only for the peat samples some data on water contents at the end of irrigation (unit?) are given in line 251.**

Our response in the online discussion: "The unit of the final water content is grams and is given in the text. Data on final water content for the peat soil and *Sphagnum* litter are available and confirm the efficacy of the irrigation treatment in causing degrees of desiccation in the treatments. This can be included in the updated manuscript."

Corrections made:

Final water weight data added to the supplement and referred to in section 2.3.

**Point 3: The different intensity of irrigation should induce different leaching rates and different DOC fluxes from the samples. No information is given on that.**

Our response in the online discussion: "Unfortunately, DOC from each irrigation event was not recorded. We note in the method section that:

> 'Previous work has shown that the amount of water used to extract DOC and whether one extraction is performed or sequential extractions to simulate multiple rainfall events gives no significant variation in DOC quality (Don and Kalbtiz, 2005, Soong et al., 2014), only changes in the total amount of carbon'.

We would therefore suggest that any differences in DOC quality are captured by our approach. Rewetting following drought of interest as this has been highlighted in the literature (and discussed in our introduction) as a period of increased riverine DOC concentrations. One of the goals of the experiments was to ascertain whether litter layer DOC flux played a role in the increased DOC concentrations post-drought or whether this was entirely due to processes within the peat, hence our focus on rewetting."

**Point 4: Following the 6 weeks of irrigation, all samples were air dried before water extraction (line 148) which does not make sense to me: If all samples were air dried before extraction, the pre irrigation to induce different degrees of desiccation seems meaningless. The rewetting of air dried soil samples cause specific effects (Birch effects) that my override the aimed irrigation effect.**

Our response in the online discussion: "Air-drying was performed so that accurate estimates of extractable DOC could be determined on a mgC $g^{-1}$ basis. Whilst air drying the samples after the irrigation simulations may have increased the homogeneity between the sample treatments, we feel this is likely to be minor as this occurred for approximately 2-3 days compared to a 42 day simulation. Also, during the simulation all samples will have been exposed to periods with no irrigation multiple times due to the number of days of rainfall being fixed across all treatments. The differences between treatments, therefore, are the extent of decomposition and DOC production during the 42 day simulation due to desiccation rather than the final water content."

**Point 5: a) The data presentation needs substantial revision: The content of tables 1 – 6 and the main message can easily be given in text form (tables 1-6 can be omitted).**

Our response in the online discussion: "We can reduce the number of tables in the manuscript if the editor feels this is necessary. Table 1 can be described easily in the text so may be removed, however Table 2 contains twelve p values as well as eight $\omega^2$ values so we feel a table is appropriate to summarise this information for the reader. It would also be possible to incorporate Tables 5 and 6 into the text if necessary."

Corrections made: Table 1, 3, 4 now described in the text.

**Point 5: b) Fig. 1 gives DOC release from the 5 sources, Fig. 2 gives drought effects on only peat samples, Fig 3 gives SUVA only for Molinia, Fig 4 gives removal efficiency for the 5 sources, but without drought effects. Hence, the presentation is confusing and inconsistent.**

Our response in the online discussion: The reasoning for this is explained in the text as treatment and/or interactive effects are interrogated. In Fig 2 drought effects were shown only for peat as this was the only DOC source with significant drought effects. Similarly, only *Molinia* was included in Fig 3 as this was the only source with a significant drought effect on SUVA. Finally, drought effects were not included in Fig 4 as there were no significant drought effects for removal efficiency.

**Point 6: The conclusions on effects of climate and vegetation change on peatland biogeochemistry are highly speculative in view of this short term laboratory study.**

Please see the online discussion for a point-by-point defence of our conclusions section. The discussions section has been significantly shortened to focus on the drought effects we observed rather than making broader comments about vegetative change in peatlands. We feel this helps differentiate this manuscript further from Ritson et al 2016 and avoids the over-interpretation the reviewer suggests.

Reviewer #2

**Point 1: The abstract states in line 29- 30 that "more immediate effects are observed in peat soils". This is correct, but if drought events will be more frequently observed in the future, these pulses of DOC can also be regarded as a long- term effect, in that they will be occurring more frequently, potentially giving a steady increase in DOC concentration.**

Corrections made:

Sentence now reads: "Long term shifts in species diversity may, therefore, be the most important impact of drought on litter layer DOC flux, whereas pulses related to drought may be observed in peat soils and are likely to become more common in the future."

**Point 2: It is somewhat surprising that drought effect was only observed with the mild treatment. This is explained by large variability in the other treatments, possibly because some samples became drier than intended (line 244- 261). The arguments are mainly repeated in lines 359- 363, but I miss a discussion of the implications of this. Do these results indicate that there is an "optimum" drought frequency for DOC release, i.e. that DOC release will not increase with increasing drought frequency and severity, but will increase to a certain point and then decline?**

Our response in the online discussion: 'Yes, we hypothesise that this is due to 'water scarcity limiting microbial activity (Toberman et al., 2008) and/or increased hydrophobic protection decreasing the extractable DOC on rewetting'. We would suggest that at very severe levels of drought DOC production is limited by water scarcity, however this would not stop oxygenation of peat and therefore greater potential for increased DOC production in the future due to the enzymatic latch mechanism. We will add a more detailed explanation of the implication of this finding to the amended manuscript.'

Corrections made:

Further discussion has been added to section 4.1 on this matter.

**Point 3: Line 423- 426: Are you suggesting that drought causes permanently altered biogeochemical controls so that the released DOM becomes gradually more aromatic? The literature usually argues that more aromatic DOM is released after single drought events, but that increased frequency of these will give increased aromaticity over time. Please explain in more detail in which way you suggest your single rewetting differs from field studies and how this may have affected the results.**

Our response in the online discussion: "In this section we discuss the lack of an increase in SUVA value for peat from the drought simulation, in contrast to field studies. The literature often suggests that DOC is elevated for many years after drought events. As we were monitoring a single rewetting event we suggest that one of the possible explanations for conflicting results could be that many of the longer term processes involved in increased DOC concentration and aromaticity (enzymatic latch, recovery from sulphate acidification from oxygenation) may not have had time to occur. We will explain this more clearly in the updated manuscript."

Corrections made:

Further discussion has been added to section 4.2 on this matter.

**Point 4: In line 431‑ 435 the results on both DOC and SUVA seem to be summarized. Do you consider that there was a "lack of drought effect for peat" or are you here only talking about SUVA? And again, you argue that the experiment simply investigates short‑ term effects. It is true, in the sense that only one single drought event is mimicked. But are there arguments that long‑ term effects of drought go beyond the sum of many single events, that there are more permanent changes going on? This is what you indicate, but you do not explain or express it clearly.**

Our response in the online discussion: "Yes, as this is in the section headed 'SUVA' we were only referring to effects on SUVA in this statement. We will clarify this in the updated manuscript. The reviewer is correct that our intention was to suggest that frequent droughts could create long term changes in peatland biogeochemistry, but that our experiment did not cover this. Again, we would be happy to clarify this in the updated manuscript."

Corrections made:

Clarified that we were referring only to SUVA and reworded concluding statement at the end of section 4.2.

**Point 5: Line 186‑ 192: Please explain why peat samples for this additional test were collected at a different site. And explain more clearly why this extra experiment was performed? Was it simply because in the main experiment there was no extraction prior to treatment, so you did this to look at changes over the course of the experiment?**

Our response in the online discussion: "The reasoning behind performing the extra experiment was to interrogate the possibility that *any* oxygenation of peat could affect DOC quantity and quality and thus explain differences between the results found here and our previous work. The reviewer is correct that this could have been achieved by extracting DOC from a sub-sample prior to the start of the original experiment, however as this was not done we performed this short experiment. The samples were collected from an ombrotrophic peatland with a comparable mixture of vegetation (*Juncus*, *Molinia*, *Sphagnum*, *Calluna*, *Eriophorum*) and were of the same level of humification (von Post scale). Although not identical to the peat collected in the original experiment, we feel these samples are similar enough to test the hypothesis that the control conditions used in the original experiment give enough oxygenation to alter DOC properties."

**Point 6: Line 439‑ 447: The discussion comes here, but it is not clear. Yes, you show that DOC removal may decline with time due to change in DOM properties, but it is not clear why this suggests that DOC removal was lower in this experiment than in Ritson et al. (2016). As far as I can see the control samples in the current experiment underwent exactly the same treatment as the peat samples in the previous experiment. Figure 4 shows DOC removal across treatments, but the results for the control group given in the supplement should be directly comparable to Figure 1 in the 2016 paper – which shows a big difference in DOC removal. I cannot see that this follow‑ up experiment explains why there is such a big difference. This is important, as you argue (e.g. in the abstract) that DOC from peat is harder to remove, but in Ritson et al. (2016) it is the easiest to remove. Please elaborate.**

Our response in the online discussion: "The confusion here lies in the experiment we are referring to in Ritson et al. (2016) as there are multiple experiments in this paper. The control group of this paper is directly comparable to the experiment entitled 'Litter decomposition in the laboratory' in the Ritson et al. 2016 paper where only data on amount of DOC extracted were presented. The comparison we were intending to make, however, is to the first experiment from the 2016 paper entitled 'Ease of DOC removal during the treatment process for different peatland sources'.

In the 2016 coagulation experiments DOC was extracted from fresh peat which had had minimal exposure to oxygen. We suggest a reason why the peat DOC in the 2017 paper showed poorer removal by coagulation was that it had been exposed to oxygen over the length of the simulation and this may have altered the treatability of the extracted DOC. The repetition of the control group conditions provides evidence for this as it shows exposure to oxygen causes a decrease in Peak C and SUVA, both of which have been correlated with ease of removal via coagulation in the literature.

We will explain this in greater detail in an updated version of the manuscript and make it clear that when we say in the abstract that that peat DOC is harder to remove we mean peat that has been exposed to oxygen compared to peat which has not."

Corrections made:

Abstract editing to clarify we mean peat DOC which has been exposed to oxygen is harder to remove. Section 4.3 expanded to explain the differences between samples in this experiment and Ritson et al. 2016 and therefore why we feel oxygenation of peat leads to DOC which is harder to remove via coagulation/flocculation.

**Point 7: Section 3.6: The fluorescence data are only presented in connection with coagulation. But what about difference in fluorescence properties related to drought treatment or vegetation type? Why are these results not presented and discussed?**

Our response in the online discussion: "These data are available and can be included in the updated manuscript. The data suggest a drought effect on Peak C (humic-like) fluorescence for both *Molinia* and *Juncus* and an effect on Peak T (protein-like) fluorescence for *Juncus*."

Corrections made:

Addition to Table 2 of ANOVA results for peak C and peak T. Section 3.3 expanded to include results from SUVA and fluorescence. Section 4.2 expanded to include discussion of both SUVA and fluorescence with the addition of the following paragraph:

'A drought effect was observed for peak C (Juncus and Molinia) and peak T (Juncus) with lower values under severe drought. These indices have been described as 'humic-like' and 'protein-like', respectively, however meaningful interpretation of the moieties responsible is difficult as many compounds can fluorescence in these regions (Aiken, 2014). From Table 5, however, we can suggest that decreases in peak C caused by drought may decrease the amenability of DOC to removal by coagulation.'

**Point 8: Line 367- 369: This probably relates to the results given in table 6, but it does not fit with the lack of drought effect on peat SUVA. I suggest just briefly mentioning this here, but refer to the lack of drought effect for peat discussed in section 4.2**

Corrections made:

Altered for clarity

**Point 9: Line 460- 462: You could mention the drought effect on SUVA for Molinia, which may partly counteract the oxygenation effect of peat (lower aromaticity).**

Corrections made:

Conclusions section has been amended to add this point in.

Technical corrections

**Line 463- 464: It is claimed that drought (oxygenation) deceases aromaticity, while the drought experiment itself did not show effects on DOM quality for peat soils. You argue why this may be so in section 4.2, but please repeat it briefly here and modify the conclusions (make them less firm).**

Done

**Line 76: You may mention what kind of programmes/how Sphagnum dominance is promoted**

Done

**Line 139: I assume the intervals between the rainfall simulation were the same, but please specify this**

Done

**Line 152: I assume the extracts were filtered before further analysis? Please explain**

Yes, added in that extracts were filtered using re-ashed GF/F filters (Whatman)

**Line 167: Was the coagulation performed on filtered samples? Incase, please justify this**

Yes, as we were working with model waters with no turbidity, filtration was performed to standardise the extracts and remove the small pieces of vegetation in the leachate.

**Line 207- 208: Move and merge into 3.1 to avoid repeating this information here**

Done

**Line 219- 222: Move the more detailed explanation of the method to section 2.4**

Done

**Line 230- 231: Specify that you are talking about the peat soil**

Done

**Line 276- 278: Specify that you are talking about the Molinia samples**

Done

**Section 3.4: Move the sentence in line 296- 297 (on drought effects) to the beginning of the section.**

Done

**Line 299- 300 simply repeats line 293- 294 – move and merge.**

Done

**Line 308- 310: Move to the discussion (section 4.2)**

Done

**Line 139: Space between "applied" and "eleven" missing.**

Corrected

**Line 151- 2: "Kalbitz" incorrectly spelled, and both references missing in the reference list. And why is the reference not put at the end of the sentence? Is the latter part the author's own interpretation?**

Citation corrected and moved to the end of the sentence.

**Line 155: Unit misspelled, should be mgC l⁻ 1.**

Corrected

**Line 266: Delete "group two Calluna and"**

Done

**Line 278: Replace "medium" with "moderate", as this is the term used elsewhere**

Done

**Line 323: Add DOC before "removal efficiency"**

Done

**Line 344+line 346 and similar places: When talking about the properties of the actual molecules in question, use DOM, not DOC. DOC is just a notation for what is actually analysed and for which we can talk about changes in concentration etc, but DOC cannot be more or less aromatic or humified.**

Done

**Line 433: Add "SUVA" before peat**

Done

**References: Sometimes access date is added, sometimes not. In general web page and access date should not be necessary for published papers, but at least be consistent.**

Apologies, this formatting was done with the automatic style for *Biogesciences* through Mendeley. The references have now been altered to be consistent (DOI, web page and access date removed).

**Line 337: I would change "without any experimental treatment" to "at control conditions"**

Done

**Line 344- 347: Delete this type of discussion text from the results chapter**

Done

Reviewer #3

**The authors do tend to over-interpret the magnitude of their results on future drought effects given this short-term laboratory study, but I still find merit in this study and recommend publication following revisions on the comments listed below.**

The discussions section has been significantly shortened to focus on the drought effects we observed rather than making broader comments about vegetative change in peatlands. We feel this helps differentiate this manuscript further from Ritson et al 2016 and avoids the over-interpretation the reviewer suggests.

Specific Comments

**Lines 38-41: Vague sentences that aren't useful to a reader as written.**

Altered to 'The extent to which conditions favourable to peat formation exist are threatened by climate change (Clark et al., 2010; Gallego-Sala and Prentice, 2012) and altered precipitation patters and more frequent droughts may also destabilise sequestered carbon (Evans and Warburton, 2010; Fenner and Freeman, 2011; Freeman et al., 2001a).'

**Line 42: "represents a significant flux of carbon from peatlands (Dinsmore et al. 2010)" - Provide a range of flux values rather than another vague sentence. Don't make the reader dig into every one of your citations to find useful information that could have easily been supplied.**

Done. Added that DOC is around 24% of NEE C uptake (Dinsmore et al. 2010).

**Line 76-77: Vague sentence. Provide useful information from this citation that explains which programmes are being promoted to increase Sphagnum dominance.**

Done. Added 'by blocking drainage ditches to re-establish high water tables'.

**Line 219-222: Description of Holm-Sidak correction should be moved to Methods section "2.4 Data analysis and statistical methods"**

Done. This was added at the first stage of revisions as in the initial submission it was queried why *Juncus* was not classed as significant in this section.

**Line 234-236: Not a result. Move to Discussion.**

Done.

**Line 245-249: Not a result. Move to Discussion.**

Done.

**Line 253-254: Not a result. Move to Discussion.**

Done.

**Line 258-261: Not a result. Move to Discussion.**

Done.

**Line 281-283: Not a result. Move to Discussion.**

Done.

**Line 316-318: Not a result. Move to Discussion.**

Done.

**Line 335-337: Do not introduce a new statistical test in the Results. Move this entire description to the Methods section "2.4 Data analysis and statistical methods"**

Citations for Sperman's, Student's and Levene's techniques added to methods section.

Technical Corrections

**Inconsistent use of "carbon" and "C" throughout the manuscript. Write "carbon (C)" the first time it is used and "C" afterwards.**

Done.

**Line 59: Change "effects" to "affects"**

Done.

---

## Author Response (AR2)

[revised manuscript text omitted]

[Reviewer comments in normal text, author response in bold]

Response to reviewer 1

While the revisions appear to address most of my initial concerns, I still found it difficult to disentangle this study from Ritson et al. 2016. I recognize that there are a number of different analyses between papers but the purpose of both papers seem nearly identical and have many overlapping results. There is a brief section in the Discussion (Lines 424-427) that highlights how Ritson 2016 differed from the current paper as Ritson 2016 extracted DOC from peat that had been exposed to a minimal amount of oxygenation during transport whereas the current study reports on peat exposed to oxygenation. I think the authors should point out this experimental difference early in the Introduction as this is an extremely important difference and the reader needs be aware of it straightaway (to avoid the same confusion I had).

**Added to introduction: "The present research sought to build on the work of Ritson et al., 2016 by assessing the effect of oxygenation of peat and vegetation due to drought on peatland DOC flux and any interaction with projected changes in litter input. The previous study had only assessed the DOC quality differences between sources collected from the field with minimal degradation/oxygenation."**

Authors highlight previous studies and define objectives of this new study, but fail to provide hypotheses regarding their choice for peatland vegetation. They do a fine enough job articulating why different plant species should be tested but without hypotheses for expected effects, the experimental design with these specific plant species remains too speculative for justification. Why did the authors choose these three specific vascular plant species? Is there a reason why these particular plant species might affect DOC differently such that they needed to be tested? Use your results from Ritson 2016 to provide hypotheses for the current study. This would also help you rationalize this study as a follow-up to Ritson 2016 and why it would make a valuable addition to the scientific community. Please include hypotheses at the end of the Introduction to address this.

**Justification of choice of species expanded in introduction so now reads: "Vegetative change in peatlands has occurred in the recent past (Chambers et al., 2007b) and is projected to continue with Sphagnum mosses, which are favoured for peat formation, giving way to vascular plants (Fenner et al., 2007; Weltzin et al., 2003). Many grassland species (Juncus effusus, Molinia caerulea) have encroached on peatland areas as a result of anthropogenic pressures such as nutrient deposition and management practices (Berendse, 1994; Chambers et al., 2007a; McCorry and Renou, 2003; Shaw et al., 1996). These species are adapted to higher nutrient availability (Aerts, 1999) and thus can out-compete peatland species if nutrient levels are elevated through, for example, nitrogen deposition (Berendse et al., 2001). Their rooting systems are also commonly deeper and more extensive than upland plant species, facilitating colonisation of peatland areas with fluctuating water tables (Lazenby 1955, Loach, 1968). Our previous work (Ritson et al., 2016) has also shown that Juncus and Molinia may be create an increase in the speed and seasonality of C cycling in peatland litter layers due to their annual cycle of production of a large amount of relatively labile aboveground biomass"**

**Wrt modifying hypothesis, the final paragraph now reads: "Previous work has highlighted both the vegetative source and climate controls on production affecting the ease of removal of DOC and the formation of DBPs (Gough et al., 2012; Reckhow et al., 2007; Ritson et al., 2014a; Tang et al., 2013). The present research sought to build on the work of Ritson et al., 2016 by assessing the effect of oxygenation of peat and vegetation due to drought on peatland DOC flux and any interaction with projected changes in litter input. The previous study had only assessed the DOC quality differences between sources collected from the field with minimal degradation/oxygenation. To this end, climate simulations of varying drought severities defined in terms of percentiles of mean monthly rainfall were performed on four typical peatland vegetation types (Calluna vulgaris, Juncus effusus, Molinia caerulea and Sphagnum spp.) and a peat soil. After a six-week drought simulation, the DOC released upon rewetting was analysed in terms of optical properties and coagulation removal efficiency with ferric sulphate to determine: (a) whether drought conditions affect DOC production from peatland litter and soil types and (b) whether**

the differences in litter quality identified in Ritson et al. (2016) between typical peatland species (Sphagnum and Calluna) and invasive, drought tolerant vegetation (Molinia and Juncus) cause different responses to drought in terms of DOC production (i.e. an interaction between the vegetation source and drought condition).

Line 145-147: A potential limitation to this study (and Ritson 2016) is the use of RO treated water to simulate rainfall. Several previous studies on rainfall simulation in peatland experiments utilize "artificial rainwater" with chemistry that matches natural rainwater. Natural (or artificial) rainwater may play a role in DOC quality that cannot be distinguished from this study. Have you considered this? **Added to Section 2.3 "RO or deionised water has been used in similar degradation studies (Cortez et al., 1996, Soong et al. 2014) and is employed so that no organic carbon is added to the samples and for the extraction step is considered to be representative of soil solutions collected** *in situ* **(Chantigny et al., 2007)."**

Line 169: Change "exiting" to "existing" **Done**

Line 334: Change "it" to "its" **Done**

Line 451-453: It appears that the authors' ultimate justification is to provide support for catchment management programmes on peatland water table levels. This justification seems to always be mentioned fleetingly and no additional information about these programmes is ever provided. Additional details (either within Intro or Discussion) about these programmes and their effectiveness at managing catchments (i.e., provide specific examples where these have been implemented) and how they can (or have) minimize(d) costs and resources at water treatment plants would be nice to see. In its current form, this justification reads more as an attempt to scale your results without provided sufficient evidence for this rationale.

**Evidence of minimised costs at treatment works are essentially the holy grail of this field of research and are, at present, lacking so hence the need for short-term studies such as this which suggest how ditch blocking may affect water treatment in the future. Many ditch blocking schemes have occurred in the UK however the timescales of blocking (<5 years) are small compared to the period of draining (typically 10-100 years) and the duration of peat formation (in the thousands of years). The introduction has been updated of include the preliminary results from the Mires project (lower peak flows in the catchment and shift to vegetation more tolerant of wet conditions).**

Response to reviewer 2

The authors have addressed my remarks to the manuscript to my satisfaction, and I recommend publishing the manuscript. My only scientific remarks is that the authors may want to report fluorescence effects on Peak C/Peak T (Line 313-319), if it adds to the understanding.
**Added to Table 1 and Section 3.3.**

Some technical remarks:
Line 41 Spelling error «patterns» **Corrected**
Line 314 "were" **Changed to 'are'**
Line 317 Capital P in peak, for consistency **All references to fluorescence peaks have been made lower case for consistency (except when at the start of a sentence)**
Table 2 and 3 Use same notation for the fluorescence variables **Done**
Line 478 "fluoresce" **Corrected**
Line 506 "decrease"? **Corrected**